# Complementary encoding of priors in monkey frontoparietal network supports a dual process of decision-making

Lalitta Suriya-Arunroj[1†‡*], Alexander Gail[1,2,3,4]

[1]Sensorimotor Group, German Primate Center - Leibniz Institute for Primate Research, Göttingen, Germany; [2]University of Göttingen, Göttingen, Germany; [3]Leibniz Science Campus Primate Cognition, Göttingen, Germany; [4]Bernstein Center for Computational Neuroscience, Göttingen, Germany

**Abstract** Prior expectations of movement instructions can promote preliminary action planning and influence choices. We investigated how action priors affect action-goal encoding in premotor and parietal cortices and if they bias subsequent free choice. Monkeys planned reaches according to visual cues that indicated relative probabilities of two possible goals. On *instructed* trials, the reach goal was determined by a secondary cue respecting these probabilities. On rarely interspersed *free-choice* trials without instruction, both goals offered equal reward. Action priors induced graded free-choice biases and graded frontoparietal motor-goal activity, complementarily in two subclasses of neurons. *Down-regulating neurons* co-encoded both possible goals and decreased opposite-to-preferred responses with decreasing prior, possibly supporting a process of choice by elimination. *Up-regulating neurons* showed increased preferred-direction responses with increasing prior, likely supporting a process of computing net likelihood. Action-selection signals emerged earliest in down-regulating neurons of premotor cortex, arguing for an initiation of selection in the frontal lobe.

DOI: https://doi.org/10.7554/eLife.47581.001

*For correspondence: lsur@pennmedicine.upenn.edu

Present address: †Department of Neuroscience, University of Pennsylvania, Philadelphia, United States; ‡Department of Otorhinolaryngology—Head and Neck Surgery, University of Pennsylvania, Philadelphia, United States

Competing interests: The authors declare that no competing interests exist.

## Introduction

We are often faced with probabilistic information guiding our decisions and actions, for example, asking ourselves which way to aim a penalty kick when the goalkeeper seems prepared to jump to the right. These kinds of prior probabilities allow us to prepare the action mostly likely to become relevant, thereby helping us economize reaction times and improve accuracy (*Dorris and Munoz, 1998*; *Basso and Wurtz, 1998*; *Gold et al., 2008*; *Suriya-Arunroj and Gail, 2015*). At the same time, we need to keep track of available action alternatives, the choice set, before commitment to a choice, for example potential reach goals in action-selection tasks (*Cisek and Kalaska, 2005*; *Klaes et al., 2011*). Here, we investigate how movement-planning areas in monkey frontoparietal cortex integrate priors with choice set information.

Priors can be defined as the a priori probability of seeing a particular stimulus, instructing an associated action, or receiving a particular reward in response to this action (*Gold and Shadlen, 2007*). When action and reward probabilities are confounded, it is unclear whether the behavioral benefit of priors results from the possibility for action planning thanks to higher predictability of future events or from higher motivation elicited by events that provide higher expected reward, defined as the product of reward amount and probability (*Neumann Von et al., 1944*; *Gold and Shadlen, 2007*). Choice biases due to reward-related priors have been investigated (*Sugrue, 2004*; *Yang and Shadlen, 2007*). Much less is known about how action-related priors bias choice (*Funahashi, 2017*) when expected reward is symmetric between both options ('free' choice). What impact does our

preliminary action plan have on our penalty shot when the goalkeeper's posture neutralizes in the last moment?

We recently demonstrated in human subjects that action priors bias reward-symmetric free choices, whereas pure reward priors do not (*Suriya-Arunroj and Gail, 2015*). Based on the idea that sensorimotor decisions are achieved through competition between action plans (*Gallivan et al., 2018*; *Gallivan et al., 2015*; *Cisek, 2012*; *Cisek, 2007*; *Klaes et al., 2012*), we postulated that action priors induce behavioral bias by establishing an imbalanced action planning. In reach tasks, reach plans are reflected by the encoding of spatially defined reach endpoints ('goals') in reach-planning brain areas. An imbalance in planning should show as different response strengths, associated with different priors, for the two goals after both alternatives are revealed and before the decision is required. Here, we tested this hypothesis in rhesus monkeys and asked if action priors proportionally modulate motor-goal encoding in the frontoparietal reach-planning areas and if motor-goal encoding influences subsequent free choices. The absence of prior encoding in the motor-planning areas or the presence of prior encoding but without effect on free choices, meaning that imbalanced action planning does not translate into free-choice biases, would argue against our hypothesis.

The information about priors needs to be maintained in parallel with currently valid alternatives (choice set) until the decision is requested. As previously observed in tasks in which multiple reach goals are narrowed down to two alternatives in each trial, sustained neural co-encoding of potential motor-goal locations during planning (*Cisek and Kalaska, 2005*; *Klaes et al., 2011*), could serve maintenance of the choice set in working memory. At single-neuron level, previously reported *potential-response* cells were activated whenever one of two equipotent reach goals overlapped with the preferred direction of the neuron (*Cisek and Kalaska, 2005*; *Klaes et al., 2011*). We thus asked if priors and choice set are encoded in parallel in movement planning areas and specifically if choice-set encoding of potential-response neurons is modulated by priors.

Finally, we ask how the transition from prior-modulated movement planning to action selection (commitment) is achieved and where first in the frontoparietal circuit. The mutual roles of frontal versus parietal sensorimotor structures in action planning and decision making are still inconclusive (*Westendorff et al., 2010*; *Hanks et al., 2015*; *Siegel et al., 2015*; *Brown et al., 2007*; *Connolly et al., 2000*). Simultaneous action-selection signals in both areas would suggest common input about action-selection outcome from other brain structures or that both areas converge onto the same solution in consensus. Instead, latency differences of selection signals between areas could be indicative of a processing hierarchy during reach selection.

## Results

Two monkeys (*Macaca mulatta*; monkeys H and K) were trained to perform rule-guided center-out reaches with sequential cueing (*Figure 1*; see also our human behavioral study *Suriya-Arunroj and Gail, 2015*). On each trial, a pre-cue, pair of arrowheads, appeared at one of four cardinal directions. The pre-cue location indicated the locations of two diametrically opposed reach goals in that given trial, 90° clockwise and counter-clockwise rotation from the pre-cue. The arrowhead sizes indicated the probabilities of the potential goals being later instructed as the final valid reach. We used seven prior levels, where 6:0 and 0:6 were full-prior conditions and 3:3 was zero-prior condition. While the pre-cue was indicated in every trial, the associated probabilities only applied to the instructed trials, in which a later rule-cue indicated the single valid goal that would be rewarded upon selection. On interspersed free-choice trials, the rule-cue was neutral, indicating that both reach goals would be rewarded with equal probability, independent of the pre-cue at trial start (goalkeeper neutralizes posture in the last moment). Although containing useful prior information, the arrowhead sizes could be ignored and are uninfluential to gain the optimal reward outcome.

One idea of the design is that motor goals are dissociated from visual stimulus locations (*Crammond and Kalaska, 1994*; *Gail et al., 2009*; *Westendorff et al., 2010*), which allows investigating motor-goal encoding independent of visual target encoding, which is important in visuomotor brain areas (*Boussaoud and Wise, 1993*; *di Pellegrino and Wise, 1993*; *Crammond and Kalaska, 1994*; *Gail and Andersen, 2006*; *Kuang et al., 2015*). Second, the task design encouraged sustained preliminary planning of two alternative movements, because the two out of four locations of the potential reach-goals varied across trials and the final instruction was provided only after a variable delay (planning period). The task discouraged premature commitment to a choice before

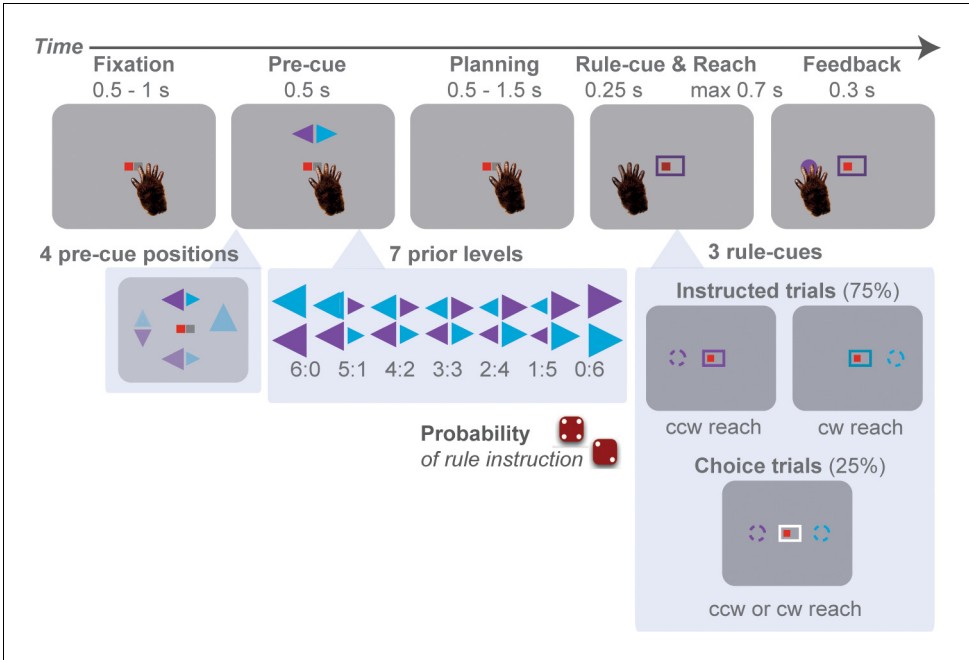

**Figure 1.** Rule-guided reach-selection task with trial-by-trial manipulation of action prior. In each trial, the monkey reached from the center of the screen to one of four cardinal directions: up, down, left, or right. The peripheral reach goals were not directly indicated by visual target stimuli. Instead, monkey inferred two potential reach locations from a double-arrow pre-cue, either clockwise (cw) or counter-clockwise (ccw) from the location of the pre-cue. Trial-by-trial, we induced variable prior by assigning probabilistic information to the sizes of the pre-cue arrowheads. The arrowhead sizes announced the prior probability with which one of the two alternative rotation rules would later be instructed by the rule-cue. A reach to the goal associated with the instructed rule (color matching the corresponding arrowhead) was always rewarded, the non-instructed (non-matching color) never. A color-neutral (white) rule-cue indicated a free-choice trial, in which both potential motor goals were rewarded equally.

DOI: https://doi.org/10.7554/eLife.47581.002

the go cue because free-choice trials were rare and the trial type, instructed or free-choice, was revealed only at the time of the go cue (*Klaes et al., 2011*; *Klaes et al., 2012*). If only free-choice trials had been offered, immediate commitment upon offering of the alternatives would have been a viable strategy. These task features allowed us to capture trial-by-trial encoding of potential motor goals during the planning period at different prior levels. Third, different to binary choice tasks, the use of four reach directions allowed estimating the spatial response properties of neurons beyond the two potential motor-goal directions. We thereby can contrast responses between cases when their coding directions align with one of the alternative reach goals (preferred and opposite directions; PD and OD) or when they do not (orthogonal directions; Orth) (*Cisek and Kalaska, 2005*; *Klaes et al., 2011*; *Glaser et al., 2018*; *Dekleva et al., 2018*).

## Biasing effects of action priors on monkeys' behavior

In instructed trials, monkeys made fewer errors when the rule-cue followed the high-prior direction and more errors when it was against the prior (*Figure 2a*). This result was indicated by the increase in error rates of *against* reaches and the decrease in *follow* reaches as a function of prior (*against*: *t*-statistic = 53.92, p<0.001; *follow*: *t*-statistic = −21.26, p<0.001; GLMM). Also, the monkeys had slower responses in *against* trials and faster responses in *follow* trials (*against*: *t*-statistic = 24.02, p<0.001; *follow*: *t*-statistic = −76.31, p<0.001; GLMM; *Figure 2b*).

   In free-choice trials, monkeys showed a choice bias toward the reach direction pre-cued with higher prior and the bias gradually increased with prior (*t*-statistic = 8.17, p<0.001; GLMM; *Figure 2c*). At the strongest prior level, monkeys almost exclusively (89%) chose the high-prior direction. Like in instructed reaches, choice RTs were slower in *against* trials (*t*-statistic = 14.61, p<0.001;

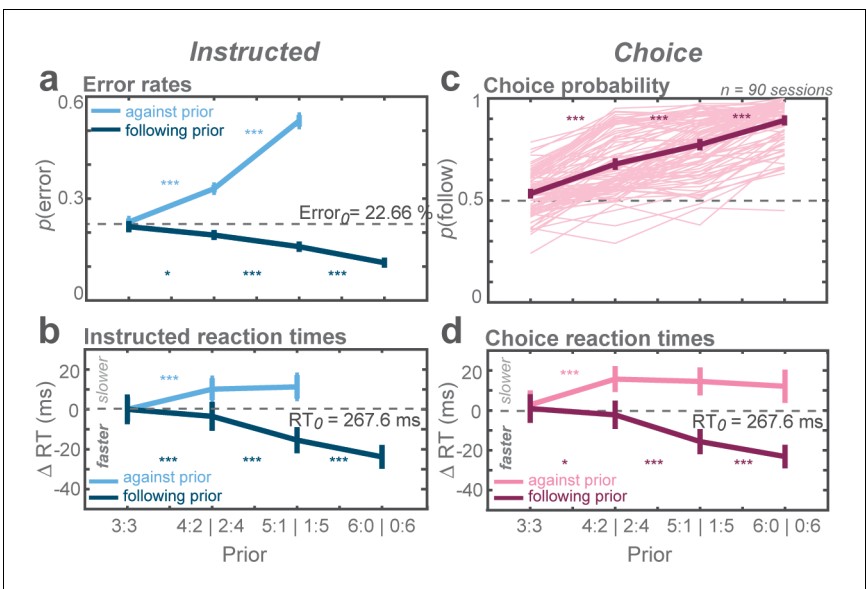

**Figure 2.** Biasing effects of action priors on the monkeys' behavior. (a) Average error rates, (b) average reaction time (RT) difference, both for trials instructing the monkey to reach the more (follow) and less (against) likely goal. *Cw* trials were arbitrarily defined as *follow* trials in the zero-prior condition. The dashed horizontal line indicates average value of both *cw* and *ccw* trials in the zero-prior instructed condition. (c) Average choice probabilities, (d) average choice RTs. Error bars depict standard errors across trials. Asterisks next to the curves indicate significant difference between neighboring data points (*$p < \alpha_{corr}$ at 5%, **$p < \alpha_{corr}$ at 1%, ***$p < \alpha_{corr}$ at 0.1%; Bonferroni-corrected *t*-test).

DOI: https://doi.org/10.7554/eLife.47581.003

The following source data is available for figure 2:

**Source data 1.** Mean error rates, choice probabilities, and reaction times.

DOI: https://doi.org/10.7554/eLife.47581.004

GLMM) and faster in *follow* trials (*t*-statistic = −42.76, p<0.001; GLMM) and the effect increased with strength of prior (*Figure 2d*).

The monkeys' behavior indicated that action priors were effective and biased their behavior in both trial types. In instructed trials, rule-cues that matched the high-prior option created behavioral benefits whereas opposite instructions created costs. In free-choice trials, action priors biased the monkeys' choice and RTs. These behavioral asymmetries were observed despite the symmetric expected reward at the moment of commitment, that is without a value-associated advantage of either option.

Both monkeys understood the task and showed near-perfect performance in full-prior trials (96% and 94%; monkeys K and H). We did observe slight effect of previous trials on the accuracy of the following instructed trials as well as the choice of the following free-choice trials (data not shown). This effect did not support win-stay/lose-shift strategy but rather suggested that monkeys sometimes corrected their behavior after an error trial.

## Complementary subclasses of graded modulation in individual neurons during planning

We recorded extracellular single-unit spiking activities from 561 well-isolated units in PMd of both monkeys (H: 238, K: 323) and 517 units in PRR (H: 248, K: 269), while animals performed the task. We analyzed spatial selectivity of the neuronal responses during pre-cue presentation, planning, and movement by comparing each unit's activity at four spatial pre-cue locations (visual encoding) and four reach directions (motor-goal and movement encoding) in full-prior instructed trials. 53% of PMd (297/561) and 50% of PRR (261/517) units were spatially selective for the motor goal during planning (*motor-goal neurons; Figure 3—figure supplement 1*) and were used for all following analyses. All isolated neurons were used for control analyses as indicated below.

Prior affected motor-goal neurons in two complementary ways. After receiving prior information from the pre-cue, the majority of the motor-goal neurons in PMd (60%) and particularly in PRR (76%) either showed increased planning-period responses when the higher prior coincided with the maximal responsive direction ($PD_{max}$) of the neuron (*Figure 3a*-left) or decreased when prior coincided with the opposite direction (OD) of the neuron (*Figure 3a*-right), but not both. For each neuron in PMd and PRR, we tested separately if the activity increased as the prior increased toward $PD_{max}$, and if it decreased when prior was directed away from it, with a linear model with factors *Prior level* and *Prior direction*. When plotting prior modulations at $PD_{max}$ against OD, most motor-goal neurons coalesced along the axes and not along the unity line (*Figure 3b*), suggesting that motor-goal neurons mostly showed either upregulation or downregulation, and rarely both. The means the example neurons in *Figure 3a* do not represent the opposite ends of a continuous distribution, but rather two distinct neuron classes, further confirmed from the bimodal distribution of relative modulation strengths in both brain areas (Hartigans dip test of unimodality; PMd: p < 0.005; PRR: p < 0.001; *Figure 3c*). Relative modulation strength was quantified as arctan of the ratio of $PD_{max}$ and OD modulation (angular distribution of the data points in the scatter plot of *Figure 3b*). The results suggest separate downregulation of the disregarded motor-goal option and upregulation of the expected motor-goal location driven by barely overlapping groups of neurons. We categorized these two main subclasses of neurons as upregulating (UR: modulation at $PD_{max}$ and angular coordinate $>\frac{\pi}{4}$) and downregulating (DR: modulation at OD and angular coordinate $<\frac{\pi}{4}$) neurons to allow separate treatment in further analyses.

## Graded neuronal modulation during planning in PMd and PRR populations correlates with graded choice bias

We quantified prior-induced modulation of motor-goal encoding during planning to ask how it impacts later choice. When we aligned the tuning functions of all motor-goal neurons according to their preferred direction (PD) to compute a grand-average population tuning (*Figure 4a*), three features were eminent.

First, population tuning showed gradual modulation with varying prior and, as suggested from single-unit analyses, the upregulating and downregulating neurons contributed to prior encoding in a complementary manner. If we momentarily disregard the two neuronal subclasses (*Figure 4a–b*), PMd and PRR grand-average populations showed higher activity when the higher prior was toward PD ($PD_{max}$: *PMd: p<0.001; PRR: p<0.001*) and lower activity with increasing prior away from PD (OD: *PMd: p<0.001; PRR: p<0.001*; GLMM; *Figure 4b*). The graded motor-goal encoding in the population (*Figure 4b*) mirrored the graded behavioral bias (*Figure 2*), even respecting idiosyncratic non-linear pattern of choice biases in each monkey (*Figure 4—figure supplement 1*). This similarity suggests a link between the level of motor-goal activity during planning and the behavior in subsequent choice. The neuronal subclasses contribute to either the up- or downregulating graded modulation, in accordance with their class definition (PMd-UR: $PD_{max}$: *p<0.001; OD: p>0.5; Orth: p>0.5;* PRR-UR: $PD_{max}$: *p<0.001; OD: p>0.05; Orth: p>0.05;* PMd-DR: $PD_{max}$: *p>0.05; OD: p<0.001; Orth: p>0.05;* PRR-DR: $PD_{max}$: *p>0.05; OD: p<0.001; Orth: p>0.05;* GLMM; *Figure 4c–d*).

Second, the encoding of both potential motor-goals during planning is supported by the downregulating neurons. In zero-prior trials, when both reach goals were equipotent, neuronal population responses were higher when the two potential reach goals matched their $PD_{max}$ and OD than when they were orthogonal to it (Orth). This response was evident in downregulating neurons (zero-prior; PMd-DR: $PD_{max}$-*Orth: $p_{corr}$ <0.001; OD-Orth: $p_{corr}$ <0.001;* PRR-DR: $PD_{max}$-*Orth: $p_{corr}$ <0.001; OD-Orth: $p_{corr}$ <0.001; Bonferroni*-corrected *t*-test; *Figure 4c–d*), whereas upregulating neurons, if at all, showed slightly higher activities when potential reach goals were orthogonal to their $PD_{max}$ (zero-prior; PMd-UR: *Orth*-$PD_{max}$: *$p_{corr}$ >0.05; Orth-OD: $p_{corr}$ <0.01;* PRR-UR: *Orth*-$PD_{max}$: *$p_{corr}$ >0.05; OD-Orth: $p_{corr}$ <0.05; Bonferroni*-corrected *t*-test). This means, in trials offering two potential motor goals, DR neurons were more active whenever their PD was part of the choice set, corresponding to bi-lobed tuning curves (*Figure 3a* right), whereas UR neurons were directionally selective only when their PD matched the higher prior option, corresponding to lack of tuning during zero-bias trials (*Figure 3b* left).

Third, prior-induced neuronal modulation was spatially restricted to the potential motor-goal locations considered in the current trial. No significant modulation was observed at orthogonal

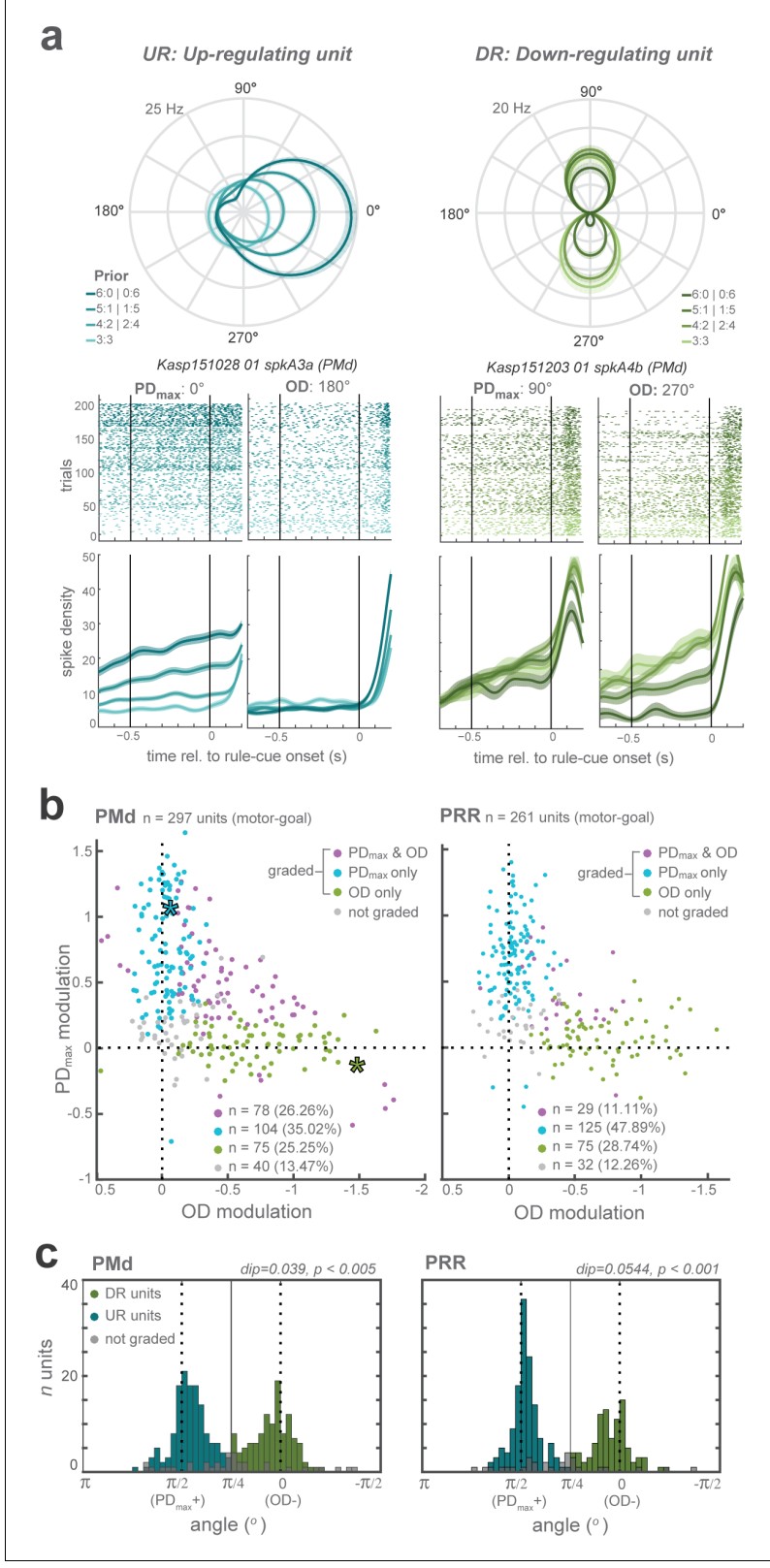

**Figure 3.** Distinct classes of prior-dependent graded modulation in single units. (a) Examples neurons with graded modulation at the neuron's maximum direction (PD$_{max}$; left panel) or opposite direction (OD; right), respectively. Polar plot tuning functions (top) and raster plots with averaged spike densities (bottom) of each unit are shown for each prior level as different shades. (b) Graded modulation is quantified as function of prior

*Figure 3 continued on next page*

*Figure 3 continued*

separately in the PD$_{max}$ and the OD, and plotted against each other, for all PMd (left) and PRR (right) neurons. The upper-right quadrant reflects increasing activities (positive modulation) of PD$_{max}$ responses and decreasing activities (negative modulation; inversed x-axis) of OD responses. Purple data points: neurons that showed significant modulation of both PD$_{max}$ and OD activities as function of prior; blue: prior-dependent modulation of PD$_{max}$ responses only; green: modulation of OD responses only; gray: no modulation. Asterisks indicate the example units shown in (a) (blue - left unit (PD$_{max}$-graded); green - right unit (OD-graded)). (c) The angular distribution of relative PD and OD modulation in all graded neurons in each area, with results of Hartigans' dip test, indicating bimodality with a concentration of the modulation along the PD$_{max}$(+) and OD(-) axes. Two shades of colors signify our henceforth two categories of neurons, up-regulating (UR) and down-regulating (DR) neurons.
DOI: https://doi.org/10.7554/eLife.47581.005

The following source data and figure supplement are available for figure 3:

**Source data 1.** Prior-modulation at PD$_{max}$ and OD, along with relative modulation strength for each PMd and PRR units.
DOI: https://doi.org/10.7554/eLife.47581.007
**Figure supplement 1.** Chamber positions, recording coordinates, and neuronal directional selectivity.
DOI: https://doi.org/10.7554/eLife.47581.006

directions (*PMd: p>0.1*; *PRR: p>0.1*; GLMM; *Figure 4a–b*). When the maximally strong prior was directed away from neurons' PD, the response strength of DR neurons even dropped below the level of Orth activities (full-prior; PMd-DR: *Orth-OD: p$_{corr}$ <0.001*; PRR-DR: *Orth-OD: p$_{corr}$ <0.001*; *Bonferroni*-corrected *t*-test; *Figure 4d*). UR neurons showed OD activity levels that were overall below the level of Orth activities and the difference became strongest at full-prior level (full-prior; PMd-UR: *Orth-OD: p$_{corr}$ <0.001*; PRR-UR: *OD-Orth: p$_{corr}$ <0.001*; *Bonferroni*-corrected *t*-test). These results suggest spatially selective enhancement and inhibition exclusively among potential motor-goal locations, while motor-goal options that have been already ruled out with the pre-cue are not modulated.

## Neuronal latencies of selection signals in PMd and PRR

Population-average spike density functions show that prior-induced modulation emerges early during pre-cue presentation then lasts until or beyond commitment to an action in both UR- and DR-neurons of PMd and PRR (*Figure 5a*). We quantify the latency of this process separately for each prior level in each neural subpopulation. To account for the dynamic transition from the planning period into the movement execution, we computed neural distances (NDs). NDs are Euclidean distances in the high-dimensional state space spanned by all neurons' responses. As proxy for the latency of an action-selection signal, we determined the time when the ND reached its maximal rate of change (maximal velocity time: MVT). Compared to thresholding average firing rates, this approach has the advantage of being applicable to complex activity increases and decreases during movement initiation and being independent of the varying starting levels between different prior conditions.

MVTs qualitatively reflected the prior-dependent RT patterns in instructed and free-choice trials (*Figure 5b–c*), being faster for *follow* versus *against* trials and scaling with strength of prior. Quantitatively, the modulation of neural latencies by prior was, however, larger than the modulation of behavioral latencies. Latency differences between brain areas are most evident in DR neurons, with PMd showing earlier MVTs than PRR (pairwise permutation test; *Figure 5c*-left). In contrast, UR neurons in both PMd and PRR showed comparable MVTs (*Figure 5c*-right). DR neurons signaled the selected movement earlier than UR neurons in PMd whereas the difference between DR and UR subclasses in PRR was not significant (Supp. *Figure 5a*). To compare all four neuronal subclasses (*Figure 5d*), we selected a subset of conditions, zero-prior condition (3:3) and one high-prior condition (5:1; shaded area in *Figure 5c*; because the number of *against* trials in full-prior conditions was limited). Despite slight variations in different task conditions, MVTs were overall reached the earliest in DR neurons of PMd (Free-choice: *against*: 210 [193–227; 90% bootstrapped CI] ms; *no-prior*: 126 [103–172] ms; *follow*:72 [53-94] ms), then in UR neurons of PMd and DR neurons of PRR, and finally in UR neurons of PRR (Free-choice: *against*: 261 [228-318] ms; *no-prior*: 204 [176-240] ms; *follow*:160 [145-190] ms; *Figure 5d*). The latest subclass, PRR-UR neurons, reached MVTs at ~50–100 ms after

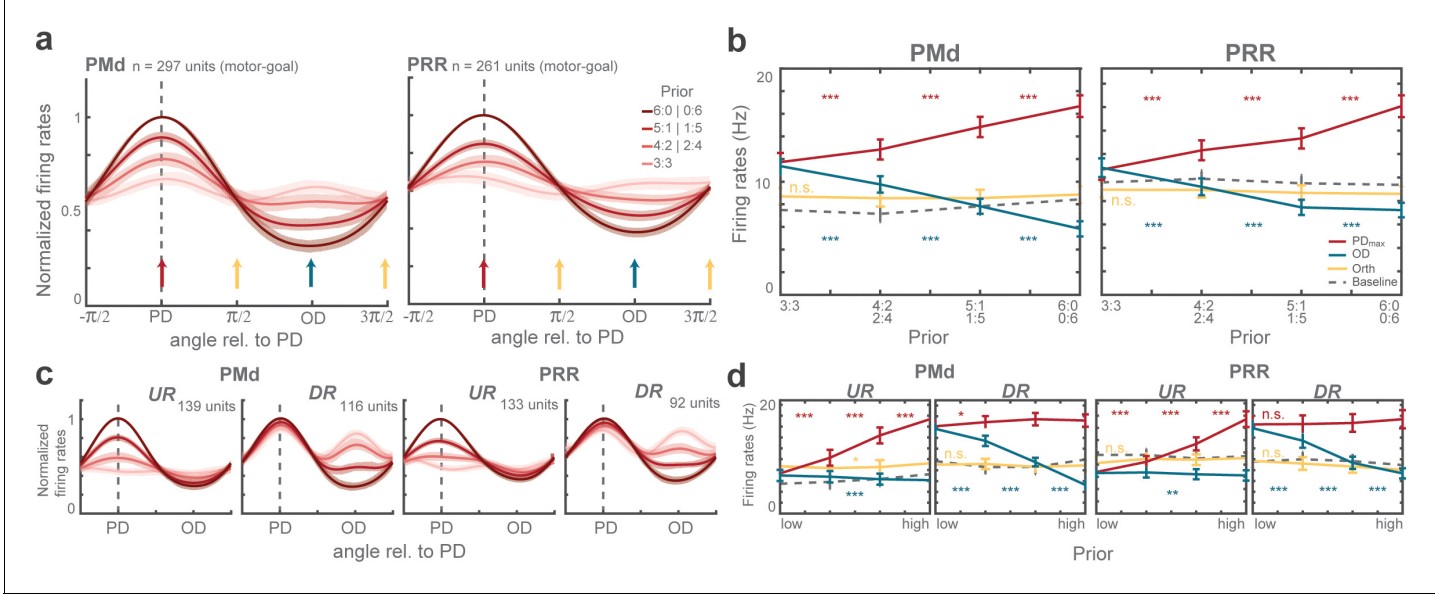

**Figure 4.** Population tuning functions and graded modulation of motor-goal activities. (a) Average normalized tuning function. Shaded areas correspond to standard errors. Vertical arrows illustrate directions quantified in (b). (b) Comparison of (non-normalized) average firing rates at maximum direction (PD$_{max}$: red), opposite direction (OD: blue), orthogonal direction (Orth: yellow) during the planning period (300 ms before the rule-cue onset), and the baseline (gray dotted line) of all motor-goal tuned PMd (left) and PRR (right) neurons. Error bars depict standard errors (*$p < \alpha_{corr}$ at 5%, **$p < \alpha_{corr}$ at 1%, ***$p < \alpha_{corr}$ at 0.1%; *Bonferroni*-corrected $t$-test). Interpolated population tunings (a) were plotted for illustrative purposes, in which we refer to interpolated firing rates and PDs continuously between 0° and 360°. Statistical analyses are based on the original neuronal activities restricted to the four discrete directions (0°, 90°, 180°, 270°) which we had sampled in our task as shown in (b). The preferred direction of a neuron was then defined by the direction toward which the motor-goal evoked the maximum response, denoted PD$_{max}$. (c–d) Same as (a–b) for upregulating (UR) and downregulating (DR) neurons separately.

DOI: https://doi.org/10.7554/eLife.47581.008

The following source data and figure supplements are available for figure 4:

**Source data 1.** Average firing rates at PD$_{max}$, OD, and Orth directions, along with interpolated tuning curve of each unit.
DOI: https://doi.org/10.7554/eLife.47581.012
**Source data 2.** Average firing rates at PD$_{max}$, OD, and Orth directions, along with interpolated tuning curve, separately for DR and UR neurons.
DOI: https://doi.org/10.7554/eLife.47581.013
**Figure supplement 1.** Planning-period activities reflect each monkey's choice bias as function of action priors.
DOI: https://doi.org/10.7554/eLife.47581.009
**Figure supplement 2.** ROC analyses of choice predictive responses.
DOI: https://doi.org/10.7554/eLife.47581.010
**Figure supplement 3.** Neuronal co-activation analysis.
DOI: https://doi.org/10.7554/eLife.47581.011

the earliest subclass, PMd-DR neurons. We performed the same analyses for plateau time (PT; when the ND stopped increasing and its velocity approached zero). PTs mostly matched movement onset or lagged behind, but overall followed the same ranking pattern as MVTs, hence showed converging results on latency differences between neuron classes and brain areas (*Figure 5b* and *Figure 5—figure supplement 1b*).

In summary, in the transition from encoding of prior-modulated motor goals to actual movement execution, neuronal signatures of action selection manifested earlier in PMd than in PRR and the leading role of PMd is driven by the DR neurons.

## Discussion

In response to trial-by-trial parametric manipulation of action priors in our rule-guided action selection task, monkeys showed gradual decreases in error rates, decreases in reaction times (RT), and increases in choice probabilities in favor of the action pre-cued with higher prior. The opposite was

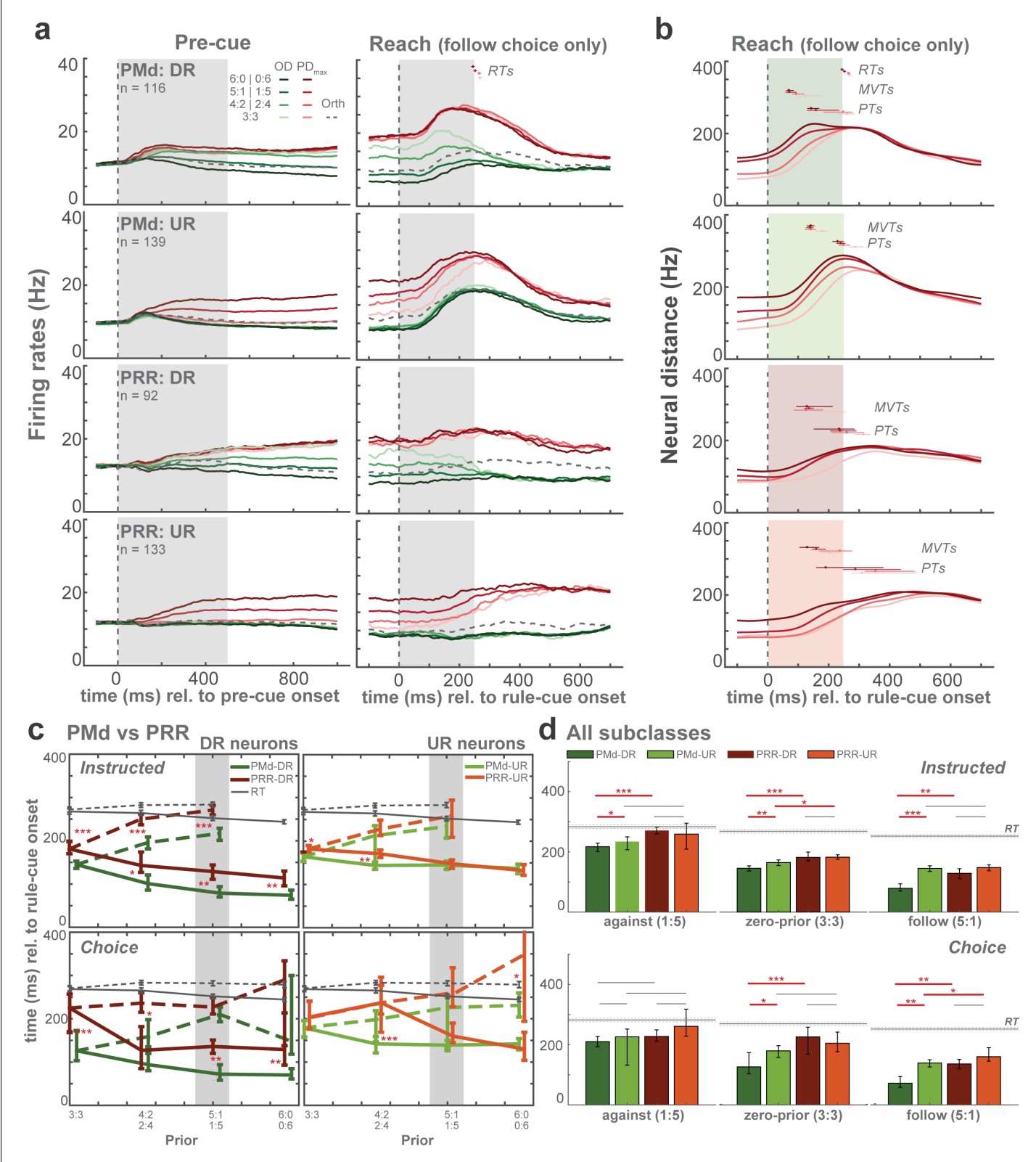

**Figure 5.** Population average of spike density functions and analyses of selection-signal latencies. (a) Average spike density functions are shown for different prior levels at maximum ($PD_{max}$: red), opposite (OD: green) and orthogonal directions (Orth: dotted gray). Orthogonal trials are only shown in zero-prior trials for better visibility. Gray-shaded areas indicate duration of pre-cue presentation in pre-cue epoch and rule-cue presentation in reach epoch. Small dots above the curves represent average reaction times (RTs) and horizontal bars represent standard errors in corresponding colors. For

*Figure 5 continued on next page*

*Figure 5 continued*

the reach epoch, only free-choice trials in which monkeys chose to follow the priors are illustrated. (**b**) Neural distances at different levels of prior aligned to rule-cue onset. RTs, maximal velocity times (MVTs), and plateau times (PTs) are shown above the curves. (**c**) Between-area comparisons of MVTs relative to rule-cue onset for DR and UR neurons. All prior conditions and both instructed (top) and free-choice (bottom) trials are shown. Solid lines indicate *follow* trials and dashed lines indicate *against* trials. RTs are also shown as gray lines. (**d**) Comparisons of MVTs. Zero-prior (3:3) and one high-prior (5:1) conditions, *against* and *follow* trials separately, are shown. Latency differences were tested between neuronal subclasses within each area (e.g. PMd-DR vs PMd-UR) and between areas only within the same neuronal subclasses (e.g. PMd-DR vs PRR-DR), depicted by horizontal gray and red lines above the bar plots (\*$p < α$ at 5%, \*\*$p < α$ at 1%, \*\*\*$p < α$ at 0.1%; permutation test).

DOI: https://doi.org/10.7554/eLife.47581.014

The following source data and figure supplement are available for figure 5:

**Source data 1.** Average spike densities functions of UR and DR neurons of PMd and PRR.
DOI: https://doi.org/10.7554/eLife.47581.016
**Source data 2.** Euclidean distances of UR and DR neurons of PMd and PRR.
DOI: https://doi.org/10.7554/eLife.47581.017
**Source data 3.** Selection-signal latencies (maximum velocity time: MTV and plateau time: PT) of UR and DR neurons of PMd and PRR.
DOI: https://doi.org/10.7554/eLife.47581.018
**Figure supplement 1.** Selection-signal latencies between DR and UR neurons of PMd and PRR.
DOI: https://doi.org/10.7554/eLife.47581.015

true for reaches against the high-prior direction. Behavioral costs and benefits of priors occurred not only on instructed but also on free-choice trials, even though both choice options offered equal expected reward at the moment the decision was required. Neuronal activity in PMd and PRR showed graded motor-goal encoding during reach planning, mirroring the graded priors and subsequent free-choice biases. Individual neurons in PMd and PRR showed prior-dependent modulation for either their preferred direction (upregulating, UR) or the opposite direction (downregulating, DR), but rarely both. DR neurons contributed to co-encoding of both potential motor goals, as seen in zero-prior trials, both in PMd and PRR. Neuronal action-selection signals were detected earlier in PMd, also driven by DR neurons. The results show how frontoparietal planning areas can integrate priors with choice sets to maintain knowledge about potential action goals while at the same time preparing the system for the most likely needed action. Latency differences indicate a potential hierarchical dependency between parietal and premotor areas.

## Graded action prior induces graded action planning, leading to graded choice bias

Our task did not involve decisions under risk. The probabilistic information contained in the pre-cue was not needed to successfully complete the trials: monkeys received an unambiguous instruction at the end of each instructed trial and were offered equal reward probability for both options in free-choice trials. Nevertheless, both monkeys showed behavioral asymmetries in instructed trials and in free-choice trials, when priors were neutralized by the white rule-cue indicating reward symmetry at the time of commitment, which we refer to as 'bias'.

Strictly speaking, our task design does not allow to distinguish action priors from reward priors during the planning period. Nevertheless, we propose that bias in our task arises from action priors. This is because the same behavioral paradigm in humans showed that only action priors but not reward priors bias free choices (*Suriya-Arunroj and Gail, 2015*). RT reductions alone, as observed in studies in which reward and action priors were confounded (*Dorris and Munoz, 1998*; *Basso and Wurtz, 1998*; *Gold et al., 2008*), are not necessarily indicative of movement preparation. Therefore, we consider the combination of instructed and prior-neutralized free-choice trials essential for contrasting the effect of action prior from reward prior. As instructed and free-choice trials in our task were indistinguishable before the 'go' instruction, the absence of a final rule instruction and the symmetric reward in free-choice trials allowed us to translate the graded cognitive state of action planning into observable graded choice behavior.

The graded action planning showed as graded neural encoding of spatial motor-goal locations in PRR and PMd. We rule out that the observed graded modulation resulted from averaging binary responses across different fractions of trials under different priors (*Dekleva et al., 2018*) by conducting two types of trial-by-trial analyses. If monkeys had prematurely committed to a choice during the

planning then planning-period activity should be indicative of the upcoming choice in each trial, including zero-prior trials. However, a receiver-operating characteristic (ROC) analysis indicated that, separately at each level of prior, the planning-period activity clearly distinguished whether higher prior was towards or away from neurons' $PD_{max}$ (*Figure 4—figure supplement 2b*) but, at each prior level, barely predicted monkeys' upcoming choices (*Figure 4—figure supplement 2c*). Further, we correlated trial-by-trial spike counts of each simultaneously recorded neuron pair which had opposite spatial selectivity ($PD_{max}$). During unambiguous encoding of a single motor goal in full-prior trials, neurons with similar PD have positive signal correlations (spike count correlations across different reach directions), neurons with opposite PD negative signal correlations, by definition. If the animals, trial-by-trial, randomly switched between both potential motor goals then these signal (anti-)correlations should be preserved even in zero-prior trials in which both potential goals were equipotent. Instead, we observed a systematic decrease in anti-correlation of neurons with opposite $PD_{max}$ with decreasing priors, suggesting co-encoding of the potential motor goals in zero-prior trials (*Figure 4—figure supplement 3*).

In summary, our trial-by-trial modulation of prior induced graded action planning which is reflected in graded motor goal encoding in PRR and PMd during sustained movement planning and leads to graded choice bias in subsequent symmetric choice.

## Complementary encoding of action priors in disjoint subclasses of neurons and dual process of decision making

Up-regulation of neuronal responses when prior was toward neurons' PD and down-regulation when prior was toward OD were supported by two hardly overlapping groups of neurons. The two complementary mechanisms for encoding priors cannot be explained by ceiling or floor effects (low-firing neurons can only upregulate; high-firing neurons only downregulate), as subsampled sets of neurons with matched firing rates were equally likely to fall into either category (data not shown; but see example neurons in *Figure 3a*). The DR neurons here likely correspond to potential response neurons described in a previous study (*Cisek and Kalaska, 2005*); the UR neurons likely correspond to a mixture of single response and build-up neurons. We did not attempt to differentiate the latter two types. As we based our classification on different criteria, namely on the pattern of prior-induced modulation, not on the bimodal tuning property, we used different nomenclature, but assume correspondence.

We ruled out that the bimodal distribution of graded units results from the preselection of motor-goal neurons in our analyses and that excluded units would fall between the DR and UR categories and restore a unimodal distribution. For this, we repeated the analysis including all recorded units in both brain areas. When analyzed in the same fashion as in *Figure 3b and 3c*, all excluded units concentrated around the origin and $\frac{\pi}{2}$ as angular measure, respectively, thereby preserving the bimodal distribution (data not shown).

DR neurons encode both potential motor goals, and are also modulated by the likelihood of the less likely goal to become instructed. Hence, they could serve as mechanism for encoding priors and choice set in parallel before commitment to a choice. DR neurons contain information about the currently valid options, in our case two out of four total. At the same time, they 'devaluate' the less likely option proportionally to the strength of prior. This could be a way of implementing choice by elimination strategies (*Tversky, 1972*), in which choice sets are reduced if factors rule out one option.

Conversely, UR neurons represent relative likelihood of choice toward their PD compared to other alternatives (*Cisek and Kalaska, 2005*) and might contribute to the encoding of decision variables (*Ratcliff et al., 2016*; *Gold and Shadlen, 2007*).

## Dual motor-goal encoding

The question whether our brain encodes dual movement goals during decision is currently still under active discussion and investigations (*Gallivan et al., 2018*). *Dekleva et al. (2018)* recently argued against dual motor representation in zero-prior two-target trials, based on the observation that monkeys showed alternating single-target selection in such trials in their experiment. First, we observed dual motor-goal encoding which is not explained by trial-averaging (ROC analysis and spike-count correlations, *Figure 4—figure supplements 1* and *2*). Second, in the same task (*Suriya-Arunroj and*

*Gail, 2015*), few human subjects verbally reported that they employed a biased strategy, focusing on one target per trial and seeing if the final instruction matched their initial plan (data not shown). Similarly, only one monkey in Dekleva et al. showed strong evidence for single reach plan representation. We thus believe that single or dual motor-goal representations can be strategy-dependent and if subjects adopt a biased strategy, one motor-goal representation will be prominent, as we also have found previously in monkeys (*Klaes et al., 2011*), but not in our current data. Third, the task design in Dekleva et al. did not encourage dual reach planning: monkeys did not have to memorize the target positions shown during the 'Target on' period as both would reappear again during the 'Go' period. It seems rather unlikely to find evidence for dual motor representation from this task. Instead it might be suited to test between single action plan vs. no plan. Task designs that require animals to retain potential motor goals in memory and in which single-trial cognitive states are sparsely and randomly probed with free-choice trials are a better common ground to compare single versus dual motor representations.

## Selective inhibition between competing motor-goal alternatives

The absence of prior-dependent modulation at orthogonal directions (Orth) and the inhibition of activities below Orth activities when prior is directed toward OD, suggest selective inhibition. Knowledge of such selectivity can help constrain computational models of biased competition in decision making. Dynamic neural field (DNF) models (*Erlhagen and Schöner, 2002*; *Cisek, 2007*; *Klaes et al., 2012*), for example, consist of computational layers that represent the continuous workspace of sensory and motor parameters, such as stimulus location or reach endpoint, and implement within-layer competition by lateral inhibitory mechanisms using center-surround excitation-inhibition (CSEI) kernels. We previously simulated biased rule-based action selection (*Klaes et al., 2012*) using a DNF model and showed bimodal motor-goal representations in the zero-prior condition that, with increasing prior, showed increasing PD activities and decreasing OD activities, while Orth responses were always lowest (*Figure 6a*). The latter contradicts our current empirical findings (*Figure 4a*).

We propose three possible explanations within the DNF framework to better account for the selective PD enhancement and OD inhibition. First, DNF can be expanded with strong mutual competition selectively between association nodes with opposing rule-preference (*cw* or *ccw*) but overlapping spatial selectivity. This corresponds to an anisotropic interaction kernel, with strongest inhibition along the 'rule' dimension in the two-dimensional association field (*Figure 6b*). We postulate corresponding association neurons to come into existence during learning of our rule-based selection task, for example, in the dorsolateral prefrontal cortex, and a strong mutual competition between them is feasible. Second, the range of mutual competition (i.e. the CSEI kernels) along the spatial dimension could be adaptive to the task, maximized at 180° apart for center-out reach task with circular workspace. In this case, a lack of modulation of Orth locations by priors could result from excitation and inhibition that cancel each other out at half-distance (in direction space) between PD and OD. Third, other decision models postulate competition between a set of discrete groups of neurons, the properties of which are defined by the specific option of the decision-making task (*Usher and McClelland, 2004*; *Brown and Heathcote, 2008*; *Ratcliff et al., 2016*; *Gold and Shadlen, 2007*), for example competition between *cw* and *ccw* goals in our experiment. Yet, we consider it important to include the implementation on a continuous coding space in the context of action-selection tasks (*Figure 6d*) to allow space-continuous motor outputs and to account for space-continuous multisensory input to posterior parietal cortex. In this sense, our suggested former two explanations provide means of achieving the third. While the first explanation suggests the main competition in the upstream association field, the second explanation requires competition mostly on the motor-goal field. A rule-guided task with varied distance or angle between reach options could help to distinguish these hypotheses.

## Neuronal selection-signal latencies suggest a hierarchy in achieving a decision

Earlier action-selection signals in frontal than parietal areas within the same animals performing the same reach-selection task suggests a frontoparietal hierarchical dependency. In contrast, saccadic choice signals during color or motion-direction categorization emerged earlier in parietal (lateral intraparietal: LIP) and dorsolateral prefrontal cortex (dlPFC) than premotor cortex (frontal eye field:

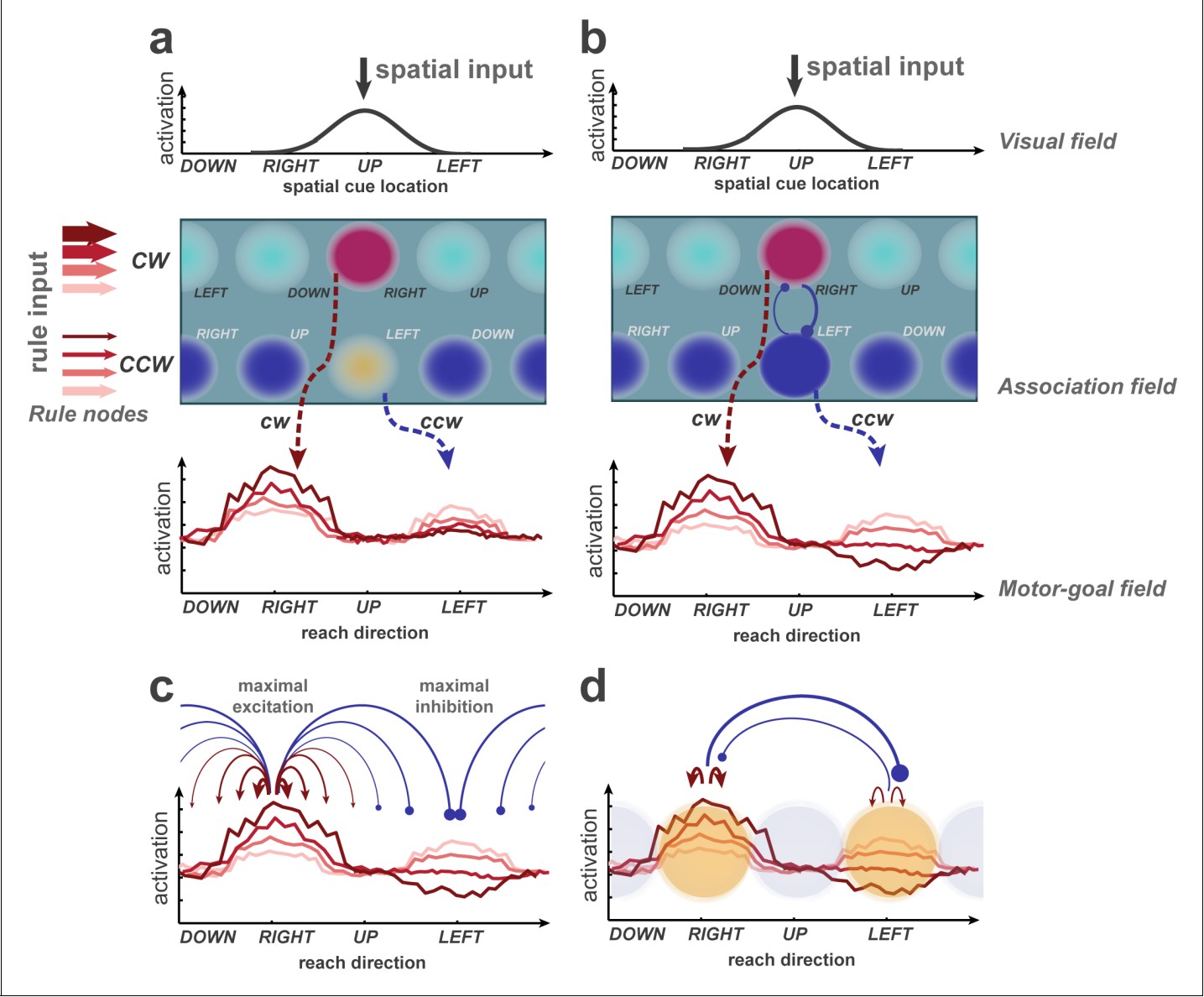

**Figure 6.** Potential explanations of selective inhibition, discussed on the basis of a learning dynamic neural field (DNF) model previously used to simulate rule-guided reach selection (***Klaes et al., 2012***). It consists of an association field, linking visual and rule inputs, and projecting onto a motor-goal field. The visual and motor-goal fields are one-dimensional, covering the space of spatial cue/reach directions. The 2D association field (AF) receives input from the visual field and two rule nodes. Putative activation levels are indicated by color temperature. Sustained representations in the AF and downstream fields are supported by local self-excitation and compete via surrounding suppression. Through training, the model learned to map a single spatial cue onto two potential motor goals in the spatial motor goal field (dashed arrows; figure conceptually illustrates simulation data from ***Klaes et al., 2012***. A rule bias enhances regions in the AF which encode the corresponding rule in conjunction with their spatial selectivity. (**b**) Hypothetical extension of the model to account for the effect of PD enhancement and OD suppression observed in the current experiment. Anisotropic lateral inhibition leads to strong mutual competition (blue circle-headed arrows) along the rule dimension, that is between regions that share the same spatial selectivity but prefer different rules. (**c**) Distance-dependent inhibition in the motor-goal field could be task-specific and maximal when the potential movement goals are 180˚ apart. (**d**) Mutual inhibition could occur selectively among currently relevant regions, representing the options, in a given decision.

DOI: https://doi.org/10.7554/eLife.47581.019

FEF) (*Siegel et al., 2015*). The discrepancy might result from differences in experimental design. First, the absence of physical target stimuli representing options in our task required the brain to construct rule-based motor goals and decide among them. Stimulus-based ('perceptual') and action-based decisions might involve different brain circuitry (*Wunderlich et al., 2009*; *Camille et al., 2011*). Second, saccade-related parietal areas might be faster than reach-related areas. In a match-to-sample categorization task requiring a manual response, average LIP activities peaked around movement onset and about 100 ms earlier than MIP (medial intraparietal; part of PRR) (*Swaminathan et al., 2013*). Late choice signals in PRR observed here and previously (*Swaminathan et al., 2013*; *Westendorff et al., 2010*) challenge a causal role of parietal lobe in rule-guided action planning and action-selection signaling. This view would agree with a previous study showing no effect on decision-making performance after LIP inactivation (*Katz et al., 2016*).

## Materials and methods

### Rule-guided reach selection task

Monkeys (monkey H/K, aged 9/11, weighing 10.34/9.80 kg, group housed) initiated a trial by directing gaze to the eye fixation spot (red square, 0.75–0.78° visual angle) monitored at 224 Hz (ET-49B, Thomas Recording, Giessen, Germany) and touching the hand fixation spot (adjacent white square) within 4.52–4.7° VA tolerance at the center of the screen (19' LCD VX922, ViewSonic, Brea, CA; IntelliTouch, ELO Touch Solutions, Milpitas, CA) 36.5–38 cm away (*Figure 1*). Each trial unfolded as follows. After 500–1000 ms fixation (random uniform distribution) the pre-cue flashed (500 ms) at one of four locations (0 °, 90°, 180°, or 270°) 8 cm (12–12.5° VA) eccentric from the screen center. Two differently colored arrowheads (magenta and cyan) of the pre-cue indicated two spatial transformation rules and hence two possible reach goals in a given trial, 90° *cw* and 90 ° *ccw* from the pre-cue, at identical eccentricity. The two pre-cue arrowheads could be differently sized, indicating the relative likelihoods of the later *cw* and *ccw* instruction. After 500–1500 ms memory period (uniform distribution), the rule-cue was presented (250 ms; 3.01–3.14° VA) as frame around the fixation spots and was either colored (magenta or cyan; instructed trials) or white (free-choice trials). In instructed trials, only the reach goal associated with the color-matching arrowhead in the pre-cue was rewarded. The association between rule and color was randomized trial by trial; monkeys had to remember and color-match the arrowheads with the later single-color rule-cue to complete the trial correctly. In free-choice trials, both potential reach goals were rewarded with equal probability. Simultaneously with the rule-cue, the hand fixation spot disappeared ('go' signal), and the monkey had to reach the goal (4.52–4.7° VA/3 cm tolerance) within a maximum of 800 ms. Successful goal acquisition resulted in a circular patch at the (chosen) goal position, a high-pitched tone, and juice reward. In failure trials, the circular patch was displayed briefly at the goal position, or both valid positions in case of free-choice trials, with a low-pitched tone, followed by 5 s timeout. Failure could be due to ocular or touch fixation breaks, misplaced or delayed reaches. Unsuccessful trials were reinserted into the trial sequence randomly. A real-time LabView program running on a PXI computer (National Instruments, Austin, TX) was used to control the tasks as well as to register all stimulus properties, event timing, and behavioral responses in each trial.

Monkey K achieved very good performance with 96% correct reaches in full-prior trials when trials were not aborted due to fixation breaks. Monkey H also understood the task very well with 94% correct reaches in full-prior trials, but high failure rate when fixation was imposed during the pre-cue. We therefore allowed monkey H to briefly break ocular fixation during the pre-cue period without the trial being aborted. Fixation had to be re-acquired before the pre-cue disappeared and maintained throughout the rest of the trial, particularly during planning and movement. Aside from stronger cue-related neuronal activities during the initial phase of pre-cue presentation in monkey H, the neuronal activity patterns of both monkeys were comparable during the rest of the trial.

### Manipulating prior

The task was designed to manipulate prior, on a trial-by-trial basis, with a pre-cue that indicated the probability that each of the two possible actions would have to be performed at the end of the trial. Instructed goals were rewarded with one unit. Either choice goal was rewarded with 100% probability but 50% chance of either receiving 1.5 units or 0.5 units, independent of pre-cue size and choice

history. We varied reward amount in free-choice trials to discourage the monkey from repeating a default selection throughout the session and encourage explorative behavior for both response alternatives instead.

The *prior* manipulation and balancing procedure were described previously (*Suriya-Arunroj and Gail, 2015*). We used seven *prior* levels, defined as likelihood ratios {6:0, 5:1, 4:2, 3:3, 2:4, 1:5, 0:6} for instruction of the *ccw* or *cw* rule, respectively. *Priors* were randomized but kept constant in blocks of eight (two free-choice, six instructed) successful trials. The ratios of initial expected values (iEV) associated with the two rules for each prior were {0.875:0.125, 0.75:0.25, 0.625:0.375, 0.5:0.5, 0.375:0.625, 0.25:0.75, 0.125:0.875}. iEVs were only valid between pre-cue and rule-cue. After the rule-cue, the EV relevant for goal selection in instructed trials became one unit for the instructed rule, zero for the non-instructed rule, and one for both rules in the free-choice trials.

We analyzed behavioral bias and neuronal modulation as function of prior, defined as absolute normalized difference between the two iEVs associated with a $prior(\frac{higher\,iEV - lower\,iEV}{higher\,iEV + lower\,iEV})$, resulting in four levels {0 0.25 0.5 0.75}. We sorted data independent of the *cw* and *ccw* rules but dependent of whether the reach was conducted toward the same (*follow*) or the opposite (*against*) direction indicated by the bigger pre-cue triangle (prior direction). We split behavioral and neural data into *follow* and *against* because preliminary analyses and our previous study (*Suriya-Arunroj and Gail, 2015*) showed asymmetric effects between both conditions. Note that *follow* and *against* responses are well-defined only in non-zero-prior conditions. In the zero-prior condition, *cw* reaches were arbitrarily considered as *follow* reaches for convenience of presentation but without relevance for the conclusions.

To encourage the monkeys to explore the possible choices, we ran a balancing task before each experimental session. The balancing task contained only zero-prior trials and, instead of rewarding both options with random probability, we used the bias-minimizing reward schedule (BMRS) (*Klaes et al., 2011*; *Suriya-Arunroj and Gail, 2015*), in which the reward probabilities reduced the more often a target was chosen in the previous free-choice trials. The balancing task was run until the monkey made at least two *cw* and two *ccw* reaches at each pre-cue position.

## Behavioral data analysis

We analyzed behavioral bias in error rate, reaction time (RT), and choice probability as functions of prior.

Error rate was the percentage of mis-placed or delayed reaches, independent of earlier trial abortion due to fixation errors. As error rates were very low and both goals were considered valid in free-choice trials, we report error rates only in instructed trials. In free-choice trials, we compute choice probabilities, defined as the fraction of correct reaches following the high-prior direction. RTs were defined as time between onset of the go-signal and release of the hand from the touch screen. RTs were corrected for monitor display and touch screen latencies.

With a generalized linear mixed model (GLMM; 'fitglme' function of MATLAB R2014b, Mathworks) we quantified influences of action priors on errors and RTs. Full models included *PRIOR* (continuous variable), *CONGRUENCY* (categorical: *follow* or *against*), and their interaction as fixed effects. *MONKEYS* were included as random effects (random slopes for *PRIOR* and *CONGRUENCY*). The likelihood of the models including or excluding different variables was compared using the MATLAB function 'compare(model1, model2)'. Our final model for error rates and RTs was:

$$X \sim PRIOR * CONGRUENCY - CONGRUENCY + (PRIOR:CONGRUENCY | MONKEYS) \qquad (M1)$$

which tests for differential effects of prior between *follow-against* responses on error rates (binomial response) and RTs with possible interactions between prior and congruency.

We tested for a biasing effect on choice using a separate full model without the *Congruency* term:

$$choice(binomial) \sim PRIOR + (PRIOR | MONKEYS) \qquad (M2)$$

Following GLMM analyses, *post-hoc* tests on error rates, RT and choice were performed to compare each pair of successive values of prior (*t*-tests, *Bonferroni* corrected for multiple comparisons).

## Animal preparation and recording procedure

All experiments complied with institutional guidelines on Animal Care and Use of the German Primate Center and with European (Directive 2010/63/EU) and German national law and regulations, and were approved by regional authorities where necessary (LAVES 3392-42502-04-13/1100).

Both monkeys were implanted with a titanium head holder custom-fit to the skull based on computer-tomographical surface reconstruction (3di GmbH, Jena, Germany) and two magnetic resonance imaging (MRI) compatible recording chambers in the left hemisphere contralateral to each monkey's dominant right hand (Horsley Clarke coordinates PRR: −12.50 /- 10.00 mm (monkey H/K) lateral; −13.50 /- 18.50 mm anterior; PMd: −19.00 /- 13.00 mm lateral, 22.00/20.00 mm anterior). Chamber placement was guided by pre-surgical and confirmed by postsurgical structural MRI, also guiding the placement of electrodes (*Figure 3—figure supplement 1*). Sustained and direction-selective neuronal responses during memory-guided center-out reach planning were used as a physiological signature in both areas to confirm the imaging-based positioning. All surgical and imaging procedures were conducted under general anesthesia and proper analgesia.

Extracellular neuronal recordings were conducted from up to five microelectrodes simultaneously in each cortical area using a five-channel Microdrive ('mini-matrix'; Thomas recording). In most sessions, simultaneous recordings were conducted in both areas. The raw signal from each electrode was pre-amplified (20×; Thomas recording), bandpass filtered, and amplified (154 Hz to 8.8 kHz; 400-800×; Plexon), while being subjected to on-line spike-sorting (Sort Client; Plexon). Spike waveforms were digitized (40 kHz) and subjected to off-line control of sorting quality and stationarity (Off-line Sorter; Plexon). All behavioral and neural data were analyzed using MATLAB and most plots were generated using GRAMM toolbox (*Morel, 2016*).

## Neural data selection and direction selectivity

All recorded units with sufficiently good isolation, stability, and activity (firing rates > 5 Hz in any trial period; see below) were included in the analysis of direction selectivity.

As known from previous studies, spatiotemporal selectivity profiles ('tuning') of individual neurons in PMd and PRR can change over time from visual-related to motor-related (*Crammond and Kalaska, 1994*; *Gail et al., 2009*; *Westendorff et al., 2010*). Neuronal spike rates were computed to reveal spatial selectivity of individual neurons Analysis time windows were: 300 ms before pre-cue onset (baseline), 300 ms after pre-cue onset (pre-cue), the last 300 ms before the rule/go-signal (planning), the time from the reach onset to offset (movement).

Directional selectivity was quantified with a directional tuning vector (DTV) (*Gail et al., 2009*; *Westendorff et al., 2010*) calculated relative to the pre-cue location (baseline and pre-cue) or reach-goal location (planning and movement). The direction of the DTV defines the preferred direction (PD) of a neuron. Significance of directional tuning was tested with a Kruskal–Wallis test *DIRECTION* as factor and sample sizes defined by the number of identical trial repetitions ($p < \sigma = 0.01$).

Basic directional selectivity was analyzed independent of *cw* and *ccw* rules in successful full-prior trials, where the definite motor-goal was announced with the pre-cue. Further analyses depended on motor-goal selectivity during planning, for which only significantly motor-goal selective neurons were used.

## Neuronal population tuning

Directional selectivity during planning was computed as function of the four motor-goal directions, separately for each value of prior. Directionally continuous tuning curves were reconstructed from the four sampled directions via ideal low-pass filtering according to the sample theorem for period signals. Each neuron was aligned relative to its PD in the planning period and normalized to the maximal response level in full-prior trials at the neuron's PD before averaging across all motor-goal neurons (*Klaes et al., 2011*) (*Figure 4a*).

## Analysis of graded neuronal modulation

We characterized graded modulation in two ways. First, at the population level, we computed the non-normalized mean firing rate of all motor-goal neurons during planning in $PD_{max}$, OD, and orthogonal trials (Orth). We tested the influence of prior using a GLMM, including *PRIOR*, *prior*

direction (*PRIORDIR*: PD$_{max}$ vs. OD), and their interaction as fix effect and individual *UNITS* as random effect:

$$Activity \sim PRIOR * PRIORDIR + (PRIOR * PRIORDIR | UNITS), \tag{M3}$$

followed by *t*-tests with *Bonferroni* correction for *post-hoc* comparisons.

Second, at the single neuron level, we assessed graded modulation by extracting trial-by-trial normalized firing rates at the PD$_{max}$ and OD and using a linear model,

$$Activity \sim PRIOR * PRIORDIR, \tag{M4}$$

to test whether neuronal activity was significantly modulated at the PD$_{max}$ and/or at the OD as a function of prior. We quantified the proportion of individual neurons that showed a significant effect in both PD$_{max}$ and OD, only in PD$_{max}$, only in OD, or no significant modulation. Additionally, we compared the strength of modulation from the model estimates (slope parameter) between PD$_{max}$ and OD. For this we tested the angular distribution (*arctan* of the ratio of PD$_{max}$ and OD slopes) for unimodality using Hartigans' dip test (*Hartigan and Hartigan, 1985*).

## Choice predictive activities and ROC (related to *Figure 4—figure supplement 1*)

For each single unit, we sorted all choice trials into PD$_{max}$ (*Prior-in*), OD (*Prior-out*), and Orth trials according to the *prior* direction (identical to *Analysis of graded neuronal modulation*). We further split the conditions into *Reach-in* and *Reach-out* choices, for reaching into versus out of the neurons' PD$_{max}$, respectively. The combinations of *Prior-in* followed by *Reach-in* and *Prior-out* followed by *Reach-out* were *follow* trials, whereas the combinations of *Prior-in* followed by *Reach-out* and *Prior-out* followed by *Reach-in* were *against* trials.

First, we tested the difference between *Reach-in* and *Reach-out* choices (*REACHDIR*) using GLMM (*Activity ~PRIOR * PRIORDIR * REACHDIR + (PRIOR * PRIORDIR * REACHDIR | UNITS)*).

Second, we tested the discriminability of choices by measuring the area under curve in a ROC analysis for each neuron separately and based on trial-by-trial planning-period activity prior to the rule-cue. The area under the ROC curve determines levels of discriminability (here: of the subsequent reach choice), where 0.5 corresponds to chance level, 1.0 to perfect discriminability. We computed the *within-condition* ROC, that is discriminability of *Reach-in* vs. *Reach-out* choices separately in *Prior-in* and *Prior-out* trials, to test how well we could predict *follow* vs. *against* choices depending on the action priors. Also, we computed the *between-condition* ROC, that is *Prior-in* vs. *Prior-out* conditions, to test how reliable we could predict choice based on *prior* directions.

## Analysis of neuronal co-activation (related to *Figure 4—figure supplement 2*)

We computed signal correlation of all pairs of motor-goal tuned neurons recorded simultaneously in the same experimental session. Between each pair, we computed pairwise Pearson correlation coefficients of the planning-period activities across all trials as a function of the distance between the neurons' PDs. The dependency of signal correlation from PD distance was computed separately for each level of prior. We used a GLMM, including PRIOR as fixed effect and neuron PAIRS as random effect (*CorrCoef ~PRIOR | PAIRS*).

### Analysis of selection signal latencies and reaction times

To illustrate the temporal dynamics of population responses, we computed spike densities *R* as function of time *t* by convolving spike trains with a causal EPSP-like kernel

$$R(t) = \frac{\tau_g - \tau_d}{\tau_d^2} * (1 - \frac{-t}{e_g^\tau}) * \frac{-t}{e_d^\tau}, \tag{E1}$$

with rise-time constant $[TMP]_{medium}g$ set to 2 ms, and decay time constant $\alpha_d$ to 50 ms. Average spike densities across trials within identical conditions were sampled at 1 ms resolution (*Westendorff et al., 2010*).

For latency analyses, we used a Gaussian kernel of width $\sigma$= 50 ms:

$$R(t) = \frac{1}{\sqrt{2\pi\sigma^2}} * e^{\frac{-t^2}{2\sigma^2}} \qquad (E2)$$

Even though not a causal filter, the Gaussian kernel allowed more reliable estimates of neural velocity (rate of change). We controlled latency results by running identical analyses on the data convolved with the EPSP-like kernel (*Westendorff et al., 2010*), yielding similar results but with less statistical power (not shown).

For assessing neural latency of commitment to a motor-goal (action-selection signal), we computed neural distance (ND). ND was defined as Euclidean distance in the high-dimensional state space spanned by all neurons, without dimensionality reduction (*Ames et al., 2014*). Because neurons were collected in different sessions, ND was computed based on the average trajectories of each task condition (congruency x prior x pre-cue location). For analysis of selection signal latency, ND was computed between trials in which monkeys reached two opposing reach goals, as function of time after the go-signal, averaged across the four pre-cue locations, separately in follow and against trials and for each prior. We estimated the variability of the ND by bootstrapping (N = 1000 resampled distances; just for illustration in *Figure 5b*). As neuronal responses differed in response to priors during planning, we could not use a non-zero ND as indicator for commitment to an action. Instead, we quantified selection signal latency as the time after the go-signal onset in which the change in ND reached maximal velocity (MVT). To confirm that the results did not depend on choice of thresholds, we also quantified the plateau latencies (PT) by calculating the time bin in which the velocity of ND became lower than 0.3 (typical velocity when the ND started to reach its peak or plateau). We did not use the maximum ND because the PRR population showed a plateau in ND rather than a peak during reach movements, which would have made peak search highly variable.

The selection-signal latencies were compared between both brain areas by pairwise permutation tests: we randomly reassigned each neuron to one of the brain areas such that the number of units in both sets matched the original sample size (N = 10,000 resampled latency differences). The percentage of random permutations leading to a latency difference larger or equal to the original sample served as the p value and when the p value was less than 5%, we considered the latency difference significant.

## Acknowledgements

We thank Nicola Popp for assisting in ROC analysis, Hao Guo for support in neural distance analyses, Cliodhna Quigley, Kyra Schapiro, Alex Filipowicz for feedback and proofreading, Sina Plümer and Klaus Heisig for technical support, Joshua Gold and Yale Cohen for helpful comments. This work was supported by the Federal Ministry for Education and Research (BMBF, Germany) grant 01GQ1005C (Bernstein Center for Computational Neuroscience) and the German Research Foundation (DFG, Germany) grants SFB-889 (Cellular Mechanisms of Sensory Processing) and GA1475-B2 (RU-1847: Physiological basis of distributed information processing for higher brain function in NHPs) awarded to AG.

## Additional information

### Funding

| Funder | Grant reference number | Author |
| --- | --- | --- |
| Bundesministerium für Bildung und Forschung | 01GQ1005C | Alexander Gail |
| Deutsche Forschungsgemeinschaft | SFB-889 | Alexander Gail |
| Deutsche Forschungsgemeinschaft | GA1475-B2 | Alexander Gail |

The funders had no role in study design, data collection and interpretation, or the decision to submit the work for publication.

## Author contributions

Lalitta Suriya-Arunroj, Conceptualization, Data curation, Software, Formal analysis, Visualization, Writing—original draft, Writing—review and editing; Alexander Gail, Conceptualization, Supervision, Funding acquisition, Writing—original draft, Writing—review and editing

## Author ORCIDs

Lalitta Suriya-Arunroj  https://orcid.org/0000-0001-8522-6778

## Ethics

Animal experimentation: All experiments complied with institutional guidelines on Animal Care and Use of the German Primate Center and with European (Directive 2010/63/EU) and German national law and regulations, and were approved by regional authorities where necessary (LAVES 3392-42502-04-13/1100). All surgical and imaging procedures were conducted under general anesthesia and proper analgesia.

## Decision letter and Author response

Decision letter https://doi.org/10.7554/eLife.47581.028
Author response https://doi.org/10.7554/eLife.47581.029

## Additional files

### Supplementary files

• Transparent reporting form
DOI: https://doi.org/10.7554/eLife.47581.020

### Data availability

All data generated or analysed during this study are included in the manuscript and supporting files. Source data files have been provided for all figures.

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
