## [Decision Letter]

Thank you for submitting your article "Complementary encoding of action priors and processing hierarchy in the frontoparietal network during decision-making" for consideration by *eLife*. Your article has been reviewed by two peer reviewers, and the evaluation has been overseen by a Reviewing Editor and Richard Ivry as the Senior Editor. The reviewers have opted to remain anonymous.

The reviewers have discussed the reviews with one another and the Reviewing Editor has drafted this decision to help you prepare a revised submission.

Summary:

The authors goal is to investigate "how action priors influence free choices and modulate motor-goal encoding in premotor and parietal cortices.

The reviewers were positively impressed by the experimental design and the results. The key findings demonstrate graded modulations that follow the "action priors". In each of the recorded areas, the authors find two populations of neurons that either are upregulated in the preferred direction or down-regulated in the opposite direction. The authors find some differences in the temporal pattern of the responses and suggest that action priors emerged earliest in down-regulating neurons of the premotor cortex.

The behavior results and the differences between populations and brain areas are potentially interesting. However, there are some concerns about the framework, some concerns about the importance of the neural results, and some concerns about the definition and interpretation of neuronal activity in terms of its behavioral correlates.

Essential revisions:

1) The manuscript heavily uses a number of terms that may be interpreted by different scientists in various ways. A most central term, "action prior" has been used by the group in other studies (and by other groups).

Action prior is basically a measure of preferring movement-feature X, which (usually) evolves by the history of the subject training in motor actions. Here, the interpretation of activity as "movement prior" is not well defined. This is especially concerning when an action prior representation is compared to other representation. The manuscript did not rule out convincingly other possibilities such as memory, reward priors, movement commitment or movement specification. It is unclear whether action prior is mutually exclusive to these other representations. The result section mixes between signals and their interpretation. Unless a clear hypothesis and definitions are provided the authors should use a more descriptive approach for presenting the result.

2) Following the above comment, this concern is especially more general. There are trends in the literature to create a new term in each paper. Science-professional literature adopted "professional terms" that are not always clear to the reader. what is exactly the meaning of the term" free choice" (is it "free"?), what is exactly "motor goal" – is it the muscles activity? Is it a type of a movement (fast/slow/ accurate/directional) ? or is it the result of an action? namely – the sensory consequences of a movement?. In this paper, it is always "rewarded action" (right?), but the goal is not defined as the reward.

3) Therefore, we suggest a more precise and careful definition of terms careful interpretation of the relation between neuronal activity and the authors favourite terms.

4) The separation to up and down regulating (UR/DR) neurons is potentially very interesting but the lack of either implantation or conceptual framework discounts the importance of these findings.

5) Additionally, to what extent is the bimodality of UR and DR neurons a product of the initial selection of neurons with significant activity? Isn't it a necessary byproduct of the initial selection of 297 PMd neurons and 261 PRR neurons (i.e., the remaining neurons would fall in between the histograms in Figure 3C)?

6) Another major concern relates to the problem in interpretation behavior of trained animal at large. What strategy did the monkeys use to solve the task? It would be important to know how trial history affected choice and RT. For example, win-stay/lose-shift could produce effects as in Figure 2. Was it only the previous trial that contributed to the effects in Figure 2, or did more trials in the past contribute? Logistic regression of choices on choice and trial history and linear regression of RT on choice and trial history could address this. If the prior was built up over many trials, it would be interesting to analyze whether the neurons' history matched that of the behavior.

---

## [Author Response]

Essential revisions:1) The manuscript heavily uses a number of terms that may be interpreted by different scientists in various ways. A most central term, "action prior" has been used by the group in other studies (and by other groups).Action prior is basically a measure of preferring movement-feature X, which (usually) evolves by the history of the subject training in motor actions. Here, the interpretation of activity as "movement prior" is not well defined. This is especially concerning when an action prior representation is compared to other representation. The manuscript did not rule out convincingly other possibilities such as memory, reward priors, movement commitment or movement specification. It is unclear whether action prior is mutually exclusive to these other representations. The Results section mixes between signals and their interpretation. Unless a clear hypothesis and definitions are provided the authors should use a more descriptive approach for presenting the result.

We agree that result descriptions should be kept free of interpretation. Throughout our manuscript we tried to further clean up and be very consistent in applying terms always with reference to the correct “level of description”. For example, we use “action prior” not for describing a preference of the subject for a certain movement (as the reviewer’s comment seems to suggest), but as the probability of a certain action to be requested from the subject as defined by the task and as indicated to the subject via the pre-cue. We call it “prior” and not “evidence”, since in the end this value will not be relevant for determining the correct choice.

Regarding dissociation from other concepts, we tried to better explain how we think that (task-level) priors translate into (behavior-level) graded action planning and choice bias, which are reflected in (neuron-level) graded motor-goal encoding. We revised the paragraph in Discussion that deals with this question. There we also explain why we consider the relevant effect here to be action prior and not reward prior. Finally, action planning is a form of prospective working memory, a fact that we address in Martinez-Vazquez and Gail 2018 Cerebral Cortex 28: 1866-1881. DOI: 10.1093/cercor/bhy035

2) Following the above comment, this concern is especially more general. There are trends in the literature to create a new term in each paper. Science-professional literature adopted "professional terms" that are not always clear to the reader. what is exactly the meaning of the term" free choice" (is it "free"?), what is exactly "motor goal" – is it the muscles activity? Is it a type of a movement (fast/slow/ accurate/directional) ? or is it the result of an action? namely – the sensory consequences of a movement?. In this paper, it is always "rewarded action" (right?), but the goal is not defined as the reward.

We added explanations to the manuscript that provide our working definitions of “free choice” (choice with symmetric reward, i.e. no evidence of either option being advantageous) and of “motor goal” (spatial location of the rewarded reach endpoint; we do not call it target, since there is no target stimulus or object at this location in our task).

3) Therefore, we suggest a more precise and careful definition of terms careful interpretation of the relation between neuronal activity and the authors favourite terms.

We cleaned up the main text and in addition to adding definitions in our revised version, we also added statements in the Discussion that should help clarify the relationship of the various concepts at the level of the task, the behavior, or the neural encoding.

4) The separation to up and down regulating (UR/DR) neurons is potentially very interesting but the lack of either implantation or conceptual framework discounts the importance of these findings.

In our revised manuscript, we suggest a dual process of decision-making which we consider relevant in situation in which multiple potential action goals (a choice set) are maintained, while at the same time priors reflect variable a priori likelihoods of the individual options. In short, DR neurons, which become active whenever their PD is part of the choice-set, and the activities of which decrease less likely the pending action matches their PD, could support choice by elimination. UR neurons, which represent relative likelihood towards their PD, could indicate net likelihood, hence support decision models that accumulate the difference between potential options.

5) Additionally, to what extent is the bimodality of UR and DR neurons a product of the initial selection of neurons with significant activity? Isn't it a necessary byproduct of the initial selection of 297 PMd neurons and 261 PRR neurons (i.e., the remaining neurons would fall in between the histograms in Figure 3C)?

We see how such concern could arise, but we carefully ruled out this possibility. We conducted the same analysis with all recorded units, including untuned units. The untuned neurons would concentrate around the origin ([0,0]) in Figure 3B and around PD_max_+ in Figure 3C, hence not fill the gap in between the two identified classes (see Author response image 1 and Author response image 2). We now added this observation (without the graphs) to the manuscript.

**Author response image 2. respfig2:** 

6) Another major concern relates to the problem in interpretation behavior of trained animal at large. What strategy did the monkeys use to solve the task? It would be important to know how trial history affected choice and RT. For example, win-stay/lose-shift could produce effects as in Figure 2. Was it only the previous trial that contributed to the effects in Figure 2, or did more trials in the past contribute? Logistic regression of choices on choice and trial history and linear regression of RT on choice and trial history could address this. If the prior was built up over many trials, it would be interesting to analyze whether the neurons' history matched that of the behavior.

Our task design discourages 1-back strategies like win-stay/lost-shift. First, we have 4 possible directions randomly interleaved, so monkeys could rarely stay at the same location over trials. Hence, the most convenient ‘stay’ was discouraged. Monkeys could only “stay” with the same rule (cw/ccw) or the same prior direction (follow/against). Second, the prior pre-cue robustly and readily biased the monkeys’ choices even in small blocks of 8 trials. Our monkeys had learned the meaning of the prior cues and their choices were biased without having to accumulate the prior information probabilistically over trials. Also, since instructed and free-choice trials were interleaved and free-choice trials were rare, we had only small fractions of trials for which we can test the effect of previous trial choices and analyze win-stay/lose-shift behavior. But we still attempted. We extracted instructed / free-choice trials that follow cw/ccw and correct/wrong trials and compare if the preceding trial had an effect. The results are compiled in Author response table 1–Author response table 4:

**Author response table 1. resptable1:** Success rate in instructed trials depending on the previous trial

		Current CW [mean ± SE; n (avg / session)]	Current CCW
Previous CW	Correct	82.15 ± 0.22; n = 198.4	72.96 ± 0.35; n = 143.8
Wrong	70.49 ± 1.41; n = 38.02	68.83 ± 2.28; n = 25.7
Previous CCW	Correct	72.14 ± 0.35; n = 145.2	81.61 ± 0.22; n = 200.0
Wrong	67.54 ± 2.29; n = 25.0	73.76 ± 1.35; n = 36.4

We do see the effect of previous trials but this slight shift doesn’t suggest that monkeys systematically followed a win-stay/lose-shift scheme. With an overall >80% performance, our monkeys did learn the task and did not just pick their choice depending on whether the previous trial was wrong or correct. We did observe that monkeys sometimes made mistakes and corrected it in the following trials, which was expected and also observed in human subjects as well.

**Author response table 2. resptable2:** Similar for follow vs. against trials:

		Current Follow [mean ± SE; n (avg / session)]	Current Against
Previous F	Correct	85.55 ± 0.12; n = 367.1	59.72 ± 0.39; n = 142.4
Wrong	80.69 ± 0.14; n = 37.5	55.83 ± 4.77; n = 13.3
Previous A	Correct	81.31 ± 0.38; n = 118.5	67.85 ± 0.93; n = 59.4
Wrong	72.7 ± 1.48; n = 37.7	62.27 ± 1.67; n = 36.7

The effect of prior still dominated. Previous trial did not reverse performance pattern; previous wrong against trials did not always lead to 100% correct against trials, which should be expected if monkeys follow lose-shift strategy. And after a wrong follow trials, monkeys didn’t get better in the following Against trials.

**Author response table 3. resptable3:** Choice probability depending on the previous trial:

		P(CW) [mean ± SE; n (avg / session)]
Previous CW	Correct	48.03 ± 0.55; n = 103.9
Wrong	48.15 ± 6.55; n = 9.73
Previous CCW	Correct	52.68 ± 0.55; n = 103.5
Wrong	48.10 ± 6.74; n = 9.6

No effect of previous trials on cw/ccw choice.

**Author response table 4. resptable4:** Choice probability depending on follow/against

		P(Follow) [mean ± SE; n (avg / session)]
Previous F	Correct	72.77 ± 0.30; n = 155.5
Wrong	67.44 ± 5.73; n = 7.8
Previous A	Correct	64.13 ± 1.04; n = 50.8
Wrong	62.18 ± 5.43; n = 11.4

Slight effect but prior effect still dominates. Also, after a wrong Against trial, monkeys did not reverse and pick more follow trials.

In summary, we did observe some small effect of previous trials (in cw/ccw instructed trials) but this effect can result from monkeys sometimes correcting their choice after an error trial but not exclusively following win-stay/lose-shift strategy.